# Moiré Attack (MA): A New Potential Risk of Screen Photos

**Dantong Niu**[1][*] **Ruohao Guo**[2] **Yisen Wang**[3,4][†]

[1]Department of EECS, University of California, Berkeley
[2]College of Information and Electrical Engineering, China Agricultural University
[3]Key Lab. of Machine Perception, School of Artificial Intelligence, Peking University
[4]Institute for Artificial Intelligence, Peking University

## Abstract

Images, captured by a camera, play a critical role in training Deep Neural Networks (DNNs). Usually, we assume the images acquired by cameras are consistent with the ones perceived by human eyes. However, due to the different physical mechanisms between human-vision and computer-vision systems, the final perceived images could be very different in some cases, for example shooting on digital monitors. In this paper, we find a special phenomenon in digital image processing, the moiré effect, that could cause unnoticed security threats to DNNs. Based on it, we propose a Moiré Attack (MA) that generates the physical-world moiré pattern adding to the images by mimicking the shooting process of digital devices. Extensive experiments demonstrate that our proposed digital Moiré Attack (MA) is a perfect camouflage for attackers to tamper with DNNs with a high success rate (100.0% for untargeted and 97.0% for targeted attack with the noise budget $\epsilon = 4$), high transferability rate across different models, and high robustness under various defenses. Furthermore, MA owns great stealthiness because the moiré effect is unavoidable due to the camera's inner physical structure, which therefore hardly attracts the awareness of humans. Our code is available at `https://github.com/Dantong88/Moire_Attack`.

## 1 Introduction

Deep Neural Networks (DNNs) present huge potential in solving varies of vision tasks such as image classification [19], instance segmentation [18], and object detection [32]. While such algorithms bring huge productivity and greatly facilitate the daily life of humans, security risks also arise. It was first revealed by Szegedy et al. [38] that small and imperceptible to human eyes perturbation on the inputs can totally fool a DNN model and dramatically alter the results [14]. Since then, large amounts of studies [14, 24, 22, 23, 10, 29, 11] have focused on crafting adversarial examples to mislead DNNs to make wrong predictions which is called adversarial attack.

In the application of DNNs, an assumption are usually assumed that image samples used in the test phase should follow the same distribution of the training data. However, in real-world applications, the image samples are usually captured by various sensors (cameras, scanners, etc.) in different conditions (light, angles, etc.). Such differences are reasonable to human eyes while may confuse a seemed robust DNN. Especially, under some situations, the strange distortions beyond the commonly used data augmentation methods may severely decrease its accuracy. One of such non-typical distortion we focus on in this paper is moiré pattern, an artifact caused by interference of overlapping lines, grids

---

[*]Work was done during an internship at Peking University.
[†]Corresponding author: Yisen Wang (yisen.wang@pku.edu.cn)

35th Conference on Neural Information Processing Systems (NeurIPS 2021).

and patterns. Moiré effect commonly occurs in the image when the frequency of the details in a scene exceeds the sensors' resolution [36]. Despite both human observers and image-capture sensors view the same image, the human perceives a normal image while colorful strange waves (moiré effect) are generated as a by-product by the sensors when the camera interacts with the frequent details of the objects (as shown in Figure 1 (a)). The moiré effect is a typical example showing such a difference between the human eyes and camera sensors. To be specific, in terms of human eyes, three different types of cone cells respond differently to light of different wavelengths, the received different color signals of the cone cells allow the brain to perceive a continuous range of colors. However, the color vision perceived by the digital sensor called color filter array (CFA) receives color discretely. The CFA is composed of many tiny color filters arranged periodically, thus the camera samples the color signals at either discrete intervals or locations. Moreover, the human observer does not even know the existence of moiré patterns for robotic observers.

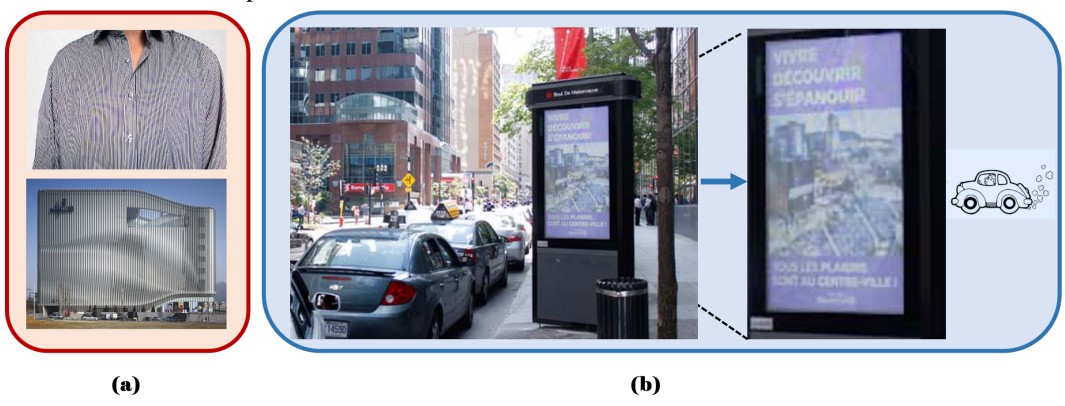

| (a) | (b) |

Figure 1: (a) Moiré pattern in the physical life; (b) Potential security risk brought by moiré pattern.

Moiré pattern disturbs the normal shooting of images and has drawn the attention of many photographers and electronic screen manufacturers. Especially, The NTIRE challenge on example-based demoiréing is organized every year to remove moiré pattern in images[3]. Simultaneously, we are wondering: *1) whether the above-mentioned difference between human eyes and image-capture sensors can be a potential physical risk to DNNs? 2) if it can be maliciously utilized by attackers to compromise the DNN-based applications?* For example, objects with frequent stripes or lines (Figure 1 (a)) may produce moiré when recorded by camera and then lead to a wrong classification. Another more common real-world scenario is illustrated in Figure 1 (b): an autonomous car is required to make real-time predictions of camera-captured images. Since the LCD monitors are now broadly existing on the streets (traffic signs, advertisements, etc.), the image captured by the sensors is highly possible to include a moiré pattern, which may trigger the subsequent false predictions of the detection and recognition system of the self-driving car, incurring a catastrophe.

In this work, we find that DNNs are vulnerable to the moiré pattern and can be easily fooled to make the false prediction (the corresponding experiments are shown in Section 3.1). This demonstrates the effectiveness of the moiré pattern as a potential physical-world attack. Then we propose a Moiré Attack (MA) by mimicking the shooting process of the camera when taking images on the LCD monitor. From the perspective of attackers, the moiré pattern can be a perfect camouflage of the adversarial perturbations. Especially, the proposed Moiré Attack is more controllable and enables the attackers to deliver the targeted attack. While from the perspective of humans or supervisors, few of them pay attention to the moiré pattern because of its unavoidability in the physical world. In summary, the contributions of our work are listed as follows:

- We first discover and analyze the potential security threat of the moiré artifacts to DNNs. Due to the mechanism of the camera's inner property to generate images, moiré pattern is unavoidable, which make the attack stealthy.

- We propose a Moiré Attack (MA) by mimicking the shooting process of the camera sensors, which gives the attackers more freedom and controllability to tamper with DNNs.

- We perform comprehensive experiments to evaluate the proposed MA, which has high attack success rate and transferability across the different victim models. In addition, it possesses a high robustness that is resistant to many extant image transformation methods.

---

[3]More information about the challenge is here: `http://www.vision.ee.ethz.ch/ntire20/`

## 2 Related Work

### 2.1 Adversarial attacks

Adversarial attack is to generate adversarial examples by maximizing the classification error of the victim model [38]. It can be applied either in a digital setting or in a physical-world setting.

**Digital attacks.** Based on the attacker's purpose, it can be divided into the targeted attack and the untargeted attack. Meanwhile, according to the information the attacker has owned, it can also be divided into white-box, black-box and gray-box attack [24]. In a white-box attack, information about the whole internal structure of the victim model is available and utilized by the attacker, most of them are combined with the optimization strategy and solved as an optimization problem. For example, represented as the basic gradient-based attack methods, FGSM and FGSM series (I-FGSM, BIM, and etc.) [14, 22, 23, 10, 29] use the gradient of classification loss with respect to the inputs as perturbations to disturb the classifiers. By contrast, in black-box attacks, attackers only know the input and final output of the victim model. In such cases, they usually construct a substitute model to simulate the victim model which is called transfer attack [38, 22, 33, 41, 40, 39] or query the victim model to generate corresponding attacks [5, 26, 2].

**Physical attacks.** In terms of physical attack, Kurakin et al. [23] first showed that, by printing and recapturing using a cell-phone camera, digital adversarial examples can still be effective. However, follow-up works had found that such attacks are not easy to realize under physical-world conditions due to viewpoint shifts, camera noise, and other natural transformations [1]. Strong physical-world attacks require large perturbations and specific adaptations over the distribution of transformations, including lighting, rotation, perspective projection, etc. Thus physical-world attacks need to generate large perturbations to increase adversarial strength, and it is always challenging to produce both effective and stealthy perturbations. Previous works proposed either controlling the perturbation into a small area or camouflaging the perturbation into target stealthy styles [3, 13, 11, 21].

In this paper, corresponding to the scenario discussed in Figure 1, we propose to use moiré pattern as adversarial perturbation intuitively. Different from previous physical attacks which achieve the adversaries by manipulating the input, the implementation of the proposed Moiré Attack is achieved by mimicking the physical shooting process of the camera. Thus unlike previous adversarial attacks aiming to conduct stealthy adversarial examples, our proposed attack based on moiré patterns is even totally unaware by human observers.

## 3 The Proposed Moiré Attack

In this section, we first investigate the moiré effect and its threat to DNNs in the physical world. Then, we formulate the problem and describe the implementation details of our proposed targeted and untargeted Moiré Attack.

### 3.1 The investigation of moiré pattern

**Moiré effect.** Moiré pattern is a large-scale interference phenomenon. It is the perception of a distinctly different third pattern caused by the inexact superimposition of two similar patterns. A moiré pattern presents different shapes (ripples, waves, and wisps of intensity variations) and frequencies when the two components move relative to each other. When the degree of their misalignment increases, the frequency of the pattern may also increase (as shown in Figure 2 (a)).

Despite almost never being seen in nature, the occurrence of the moiré pattern is ubiquitous in real photographing. When using the camera or other digital devices to capture a scene or object, if the repetitive details (such as lines, dots, etc.) within the scene exceed sensor resolution, the camera produces strange-look patterns. Moiré pattern can be easily found when photographing daily objects around us such as all kinds of fabrics (jackets, towels, shirts, and curtains), stripe-decorated architectures, various display screens and even straight hairs, as shown in Figure 1 (a). In other cases, especially when taking photos on LCDs, the moiré pattern is unavoidable because the process of displaying and capturing image is discrete in digital imaging. In detail, the digital display is composed of many pixels. The pixel grid is further divided into single-color regions for either displaying or sensing the color. As most displays and image-acquisition systems cannot display or sense the different color channels at the same site, a color is typically represented by subpixel, including several

component intensities such as red, green, and blue, in which the intensity of each pixel is a variable. The pixel (composed of subpixels) appears as a single color to the human eyes because of blurring by the optics and spatial integration by nerve cells in the eye. However, the subpixels are visible when viewed by a high-resolution camera and sometimes appear with moiré patterns. Cameras perceive the color information by CFA, which filters the light by wavelength range (as shown in Figure 2 (b)). The image perceived by CFA is called a raw image, then demosaicing is used to recover the raw image into a full-color image. In the process of image processing, the sample of color information is discrete, resulting in moiré patterns overlaid on the final image. With the variations in position of camera, types of cameras, the moiré patterns appear on the images are always diverse.

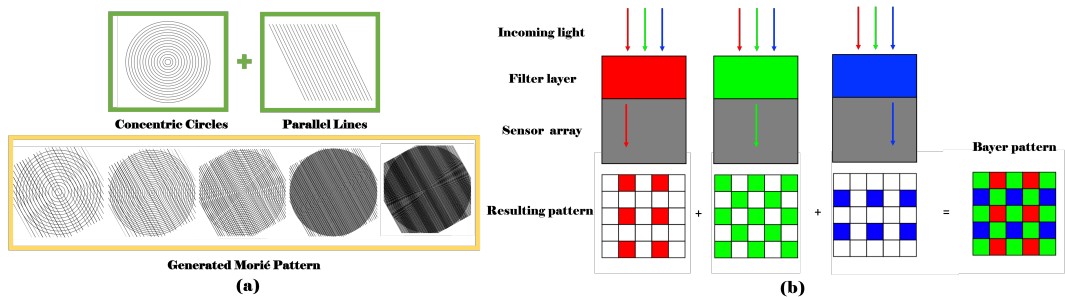

Figure 2: (a) Generation mechanism of moiré pattern. The upper line are two basic stripes, the lower are five generated moiré patterns by overlapping them with different frequencies (the frequency increases from left to right). (b) Bayer pattern used in single-chip cameras.

**Threat to DNNs.** We consider the unavoidable moiré pattern generated in the shooting may become a physical threat to DNNs that leads to wrong predictions. For further exploration, we conduct the following experiment. We adopt different LCD displays: Outdoor LCD display, Sony TV display, large display and Dell computer monitor. In terms of the cameras, we use Google Pixle 4, Sony Camera, and Huawei Pro20. For each type of digital display, we use all the cameras to take 20 photos of each. The photos are taken from several angles: $30°, 45°, 60°, 75°, 90°, 105°, 130°$, and $145°$. The photos taken on the display by the camera and the original digital images are both feed into DNNs to make predictions. Some of the results are shown in Figure 3. It indicates that moiré pattern can be a potential security risk in real-world scenarios when we use digital devices to take images on LCD monitors. The irregular pattern may trigger DNNs to make false predictions or be maliciously utilized by attackers as a camouflage to deliver digital moiré attacks. In the next part, we try to use the moiré pattern to further craft the digital attack by implementing its generation process.

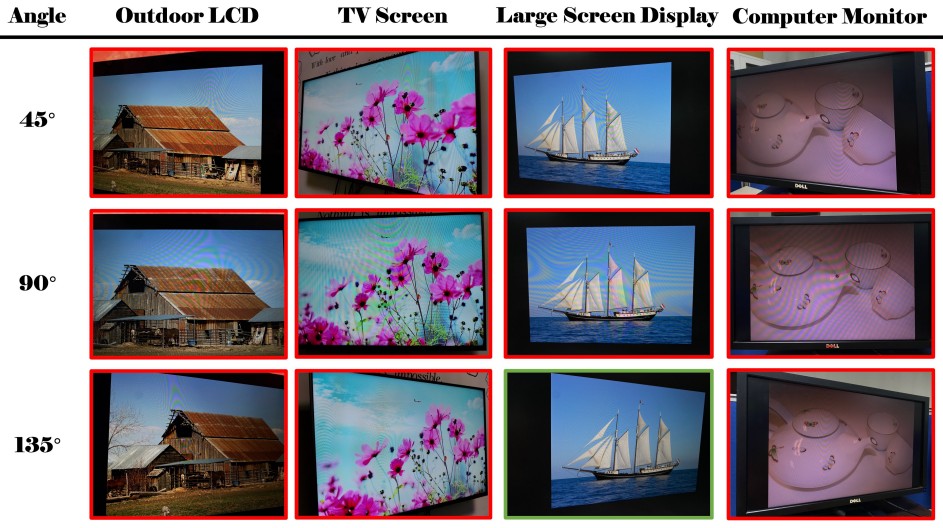

Figure 3: The threat of moiré pattern to DNNs. Red frame represents a successful attack where moiré pattern triggers DNN to make false prediction, while the green frame represents the failed one.

## 3.2 The formulation of Moiré Attack

**Preliminaries.** Given a clean image $x \in \mathbb{R}^d$ with the ground truth $y_{truth}$, a DNN classifier $f : \mathbb{R}^d \to \{1, \cdots, k\}$ which maps the image data to a discrete label set of $k$ classes, and a target class $y_{adv} \neq y_{truth}$ for the targeted attack. The goal is to find an adversarial example $x^*$ for the clean image $x$ that leads to $f(x^*) \neq y_{truth}$ or $f(x^*) = y_{adv}$. Typically, $x^*$ is restricted by $L_p$ norm: $\|x^* - x\|_p \leq \epsilon$, where $\epsilon$ denotes the perturbation budget.

In this work, we generate the adversarial examples by mimicking the shooting process of a camera to take photos on LCD. The process can be seen as a differentiable transformation that we denote as $g(x, \delta, \kappa)$, where $\delta$ is the sensor noise, $\kappa$ denotes the other parameters. The goal of our digital Moiré Attack is to optimize the sensor noise $\delta$ to craft the adversarial images that leads $f(g(x, \delta, \kappa)) \neq y_{truth}$ (untargeted attack) or $f(g(x, \delta, \kappa)) = y_{adv}$ (targeted attack), where $x^* = g(x, \delta, \kappa)$. It should be noted that different from most of the digital adversarial attack methods, the moiré pattern is an unavoidable physical phenomenon, which means people tend to consider it reasonable even the moiré distortion is relatively large. Based on this, the $L_p$ norm restriction is unnecessary for the generated examples with the moiré pattern. However, for comparison, we still use $L_\infty$ norm in our attack process and finally find that the proposed Moiré Attack is budget-free, which is a great property of the Moiré Attack.

As mentioned above, the aim of Moiré Attack is to find a sensor noise $\delta$ during the shooting process so that induces the classifier to make the wrong predictions. To solve the problem, we reformulate it as an optimization problem, and our objective is:

$$\min_{\delta} \; \mathcal{L}_{adv}(x^*, y_{truth}), \text{ where } x^* = g(x, \delta, \kappa)$$
$$\text{s.t. } \|\delta\|_\infty \leq \epsilon, \tag{1}$$

where $g(x, \delta, \kappa)$ denotes the differentiable process of simulating the generation of moiré pattern, $\delta$ is the sensor noise of the digital image-capture device. We calculate the gradients during the backward propagation to find the suitable $\delta$ to craft the adversarial examples. The pipeline of the proposed Moiré Attack is shown in Figure 4.

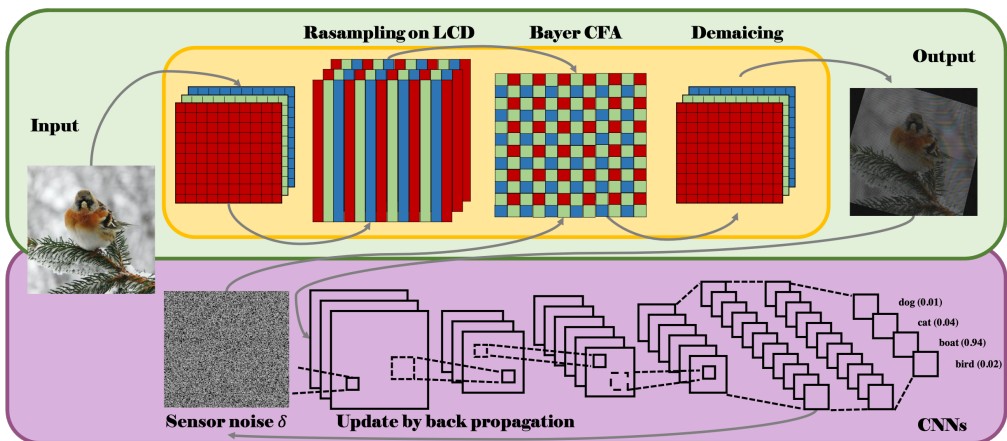

Figure 4: The pipeline of the proposed Moiré Attack (MA). To generate the moiré pattern, the input image is first fed into the yellow module which mimics the shooting process of taking photos on LCD monitors in which an iterative sensor noise $\delta$ is added in the step of Bayer CFA. The output image with moiré pattern is then fed into a CNN to make a prediction where backpropagation is conducted to iterate the sensor noise $\delta$ that misleads the CNN to make wrong predictions.

**Adversarial loss.** We use cross-entropy loss as the adversarial loss $\mathcal{L}_{adv}(\cdot)$ for Moiré Attack, it is defined as:

$$\mathcal{L}_{adv} = \begin{cases} \log(p_y(x^*)), & \text{for untargeted attack,} \\ -\log(p_{y_{adv}}(x^*)), & \text{for targeted attack} \end{cases} \tag{2}$$

where $p(\cdot)$ is the probability output (softmax on logits) of the target model $f$ with respect to class $y_{adv}$ or $y_{truth}$. By minimizing loss $\mathcal{L}_{adv}$, Moiré Attack finds an optimal sensor noise $\delta$ to simulate the image capturing process on the LCD of digital devices.

### 3.3 The implementation of Moiré Attack.

In our Moiré Attack, we strictly follow the process of the image display on a LCD and the pipeline of optical image capture and digital processing of the camera or smart phone. The pipeline is similar to [27, 44] and can be summarized in the following steps. The complete pseudo-code for Morié Attack is shown in Appendix A.

**1) Resize the image to the size of the LCD monitor.** Before an image is displayed on an LCD monitor, it should be resized to a suitable size to cater to the resolution of the LCD device. In our paper, we denote the size transformation parameter as $\alpha$. In Section 4.5, we further explore the relationship of the generated moiré pattern and the parameter $\alpha$.

**2) Resample the input RGB image into a mosaic RGB subpixels.** Image displayed on the LCD is spatially resampled to subpixels. In our experiment, we model the RGB pixels as 9 subpixels [R, G, B; R, G, B; R, G, B]. It is shown that this step causes the final images to be darker.

**3) Apply random projective transformation on the image.** In order to simulate the different relative positions and orientations when capturing an image on the LCD, we rotate the image displayed on the LCD with a random angle $\gamma$, in our experiment, $\gamma$ is in the range from $-45°$ to $45°$. In addition, the radial distortion function is used to simulate the lens distortion.

**4) Resample the image using bayer CFA to simulate the raw reading of the camera sensor.** In digital imaging, a color filter array (CFA) or color filter mosaic (CFM) is a mosaic of tiny color filters placed over the pixel sensors of an image sensor to capture color information [30]. The most commonly used CFA is Bayer CFA (as shown in Figure 4). The Bayer pattern (also called RGGB filter) is in a size of $2 \times 2$ and measures the green image on a quincunx grid (half of the image resolution) and the red and blue images on rectangular grids (quarter of the image resolution) [25].

**5) Add the perturbation to simulate the senor noise.** To simulate the sensor noise, Gaussian noise is used in previous work [27, 44]. Here we denote the sensor noise as $\delta$ with a $L_\infty$ budget $\epsilon$ and optimize it through the back propagation of the DNNs.

**6) Apply demosaicing and denoising.** Demosacing is the inverse processing of mosaicing. It is used to reconstruct a full-color image from the incomplete color sample output from an image sensor overlaid with a color filter array (CFA). To reconstruct the image from the data collected by the CFA, we use bilinear interpolation in this work. Finally, we denoise the image with the standard denoising function provided by OpenCV. Until now, an image with the moiré pattern is generated.

## 4 Experiments and Evaluation

### 4.1 Experiment setup

The proposed Moiré Attack is implemented by Pytorch on the NVIDIA Tesla V100 GPU.

**Dataset.** Since ImageNet is a large and comprehensive dataset, we conduct experiment on **ImageNet** [9] validation dataset and randomly select 5000 images which can be correctly classified by the victim model as the clean examples.

**Victim model.** We use Inception-V3 [37] as the victim model for all the experiments. To analyze the transferability of Moiré Attack, we use Resnet34 [19], Resnet50 [19], Densenet121 [20], VGG16 [35] and VGG19 [35] to evaluate the generated adversarial examples.

**Baselines.** To evaluate the robustness of Moiré Attack, different defense methods (Image Transformation (IT) [15], JPEG compression [34], Pixel Deflection [31], Feature Squeezing [42]) are adopted. For comparison, we compare our proposed Moiré Attack with: PGD [28], BIM [23], and MI-FGSM [10] under the $L_\infty$ setting, and C&W [4] under the $L_2$ setting.

**Metrics.** We use the success rate ($Succ$ (%)) to evaluate the attack ability of our method, which is the proportion of the successful attacks among the total number of test images.

## 4.2 The effectiveness of Moiré Attack

To evaluate the effectiveness of the proposed Moiré Attack, we conduct the targeted and untargeted Moiré Attack. To exclude the effect of rotation and dim light, we also test the rotated and dim images under the same setting. The generated examples and details are shown in Figure 5 and the attack sucessful rate is shown in Table 1. From the results, we can see the processing of rotation and dimming hardly change the predictions of the victim model, which means DNNs are relatively robust against such common transformations. For targeted attack. the generated adversarial examples are shown in Figure 5 (a) (the $\epsilon$ shown in the figure is set to 4). The targeted attack class is set to "hen". From the results, the success rate can reach $97.0\%$ when $\epsilon = 4$. With the increase of $\epsilon$, the success rate of attack increases. For untargeted attack, the generated adversarial examples are shown in Figure 5 (b) (the $\epsilon$ shown in the figure is set to 2), the success rate can reach $100\%$ with a small noise budget. In the following ablation study (Section 4.5), we further explore the influence of the noise budget $\epsilon$ in the perspective of perception to human eyes, which presents another great property of the proposed Moiré Attack - the budget-free sensor noise can be a perfect camouflage of the attack. From Figure 5, we can see that the generated adversarial example is visually natural in the sense of an physical attack, its visual rationality can be confirmed by the comparison with other physical attacks in Appendix B.

Table 1: Success rate of Moiré Attack.

| Attack | Rotate | Rotate + Dim | Targeted MA | | | | Untargeted MA | | | |
|--------|--------|--------------|-------|-------|-------|-------|-------|-------|-------|-------|
| | | | 2 | 4 | 6 | 8 | 2 | 4 | 6 | 8 |
| Succ | 0.122 | 0.163 | 0.876 | 0.970 | 0.990 | 0.995 | 1.000 | 1.000 | 1.000 | 1.000 |

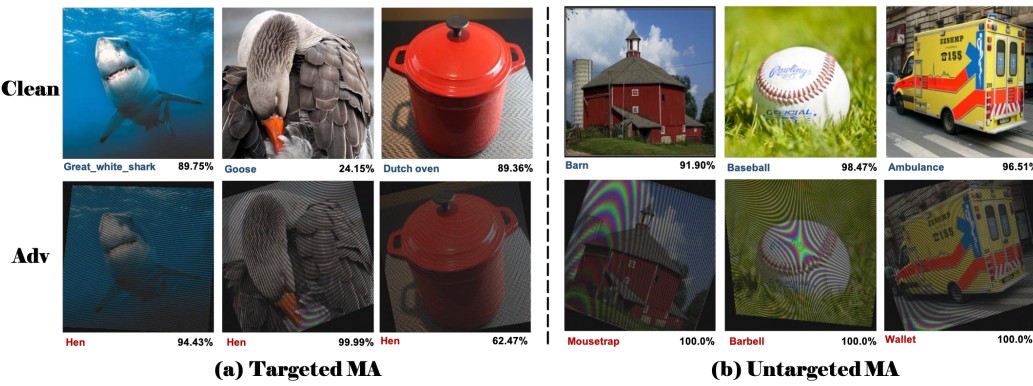

Figure 5: Adversarial examples generated by targeted (a) and untargeted (b) Moiré Attack.

## 4.3 Robustness analysis

**Under simple median value normalization.** During the real shooting process, the shoot image with moiré pattern is darker than the original digital images (the rationality of the darkness is discussed in Appendix C). In our experiment, we perform mean value normalization to brighten the generated adversarial example and then re-test the success rate of the Moiré Attack. We set the mean value of the final adversarial examples the same as the clean image by subtracting their difference. We set the noise budget as 2, 4, 6, and 8. The attack success rate is shown in Table 2 , we can see MA is relatively robust to such a mean value processing.

Table 2: The success rate of untargeted MA after mean value normalizing.

| Attack | Untargeted MA | | | |
|--------|-------|-------|-------|-------|
| Noise budget $\epsilon$ | 2 | 4 | 6 | 8 |
| Success rate (mean value normalized) | 0.960 | 0.987 | 0.994 | 0.998 |

**Resistant to JPEG compression.** JPEG compression has been revealed by many studies to be an effective defense against the adversarial attack [12, 7, 8, 34], as the added small perturbation can be removed in the process of compression. In JPEG compression, the quantifiable quality $qf$ decides the degree of the compression (smaller $qf$ means more information are reduced). In our experiment, we test the success rate of Moiré Attack under JPEG compression with different $qf$ and make a comparison with the baselines. The success rate is shown in Table 3. From the results, we can see that Moiré Attack is very robust against JPEG compression. Even the quantifiable quality $qf$ is 20 that is a relatively low value, the attack success rate can still remain a high value 93.8% while other baselines are subject to such compression.

Table 3: Success rate of Moiré Attack under the JPEG compression. The second left column is the attack success rate on Inception-V3 model, the right 4 columns are the corresponding attack success rate under the JPEG compression with different $qf$.

| | Inception-V3 | $qf = 20$ | $qf = 40$ | $qf = 60$ | $qf = 80$ |
|---|---|---|---|---|---|
| MA | **1.000** | **0.938** | **0.949** | **0.956** | **0.9881** |
| BIM ($L_\infty$) | 0.989 | 0.286 | 0.431 | 0.606 | 0.853 |
| PGD ($L_\infty$) | 0.99 | 0.276 | 0.379 | 0.522 | 0.797 |
| CW ($L_\infty$) | 0.762 | 0.205 | 0.207 | 0.226 | 0.354 |
| MI-FGSM ($L_2$) | 0.991 | 0.491 | 0.776 | 0.892 | 0.964 |

**Under other image transformation methods.** In this part, we evaluate Moiré Attack under several other transformations and denoise methods: pixel deflection (PD) [31], feature squeezing [42], and total variance minimization (tvm) and image quilting (image transformation (IT) methods) [15]. All the parameters are set following their original papers. We set noise budget $\epsilon = 4$. All the attacks are untargeted attacks. The results are shown in Table 4, we can see the proposed Moiré Attack is relatively stable compared with other baselines and can still retain high success rate under varies image transformation methods.

Table 4: Attack success rate under different transformation methods. The second left column shows the attack success rate on victim model Inception-V3 without transformations.

| Attacks | Inception-V3 | PD | Feature Squeezing | | | | IT | |
|---|---|---|---|---|---|---|---|---|
| | | | bit depth | | median smoothing | | tvm | quilting |
| | | | 4 | 5 | 2 * 2 | 3 * 3 | | |
| MA | **1.000** | **0.988** | **0.998** | **1.000** | 0.848 | 0.880 | **0.985** | **0.978** |
| BIM | 0.989 | 0.714 | 0.930 | 0.984 | 0.918 | 0.850 | 0.810 | 0.530 |
| PGD | 0.990 | 0.661 | 0.945 | 0.984 | 0.890 | 0.796 | 0.812 | 0.522 |
| CW | 0.762 | 0.109 | 0.062 | 0.182 | 0.074 | 0.121 | 0.804 | 0.516 |
| MI-FGSM | 0.991 | 0.931 | 0.976 | 0.988 | **0.962** | **0.945** | 0.820 | 0.557 |

**Under demoiréing methods.** We further explore whether Moiré Attack will be affected by the extant demoiréing methods. We search for the challenge of demoiréing [43, 44] in recent years and choose four SOTA methods MopNet [16], MDDN [6], FHde2Net [17], and AMNet [45] to test MA (the settings are the same as the ones in their papers). We show the success rate of untargeted MA ($\epsilon = 4$) under demoiréing methods in Table 5 and the visualization of demoiréing results in Appendix D.

Table 5: The success rate of untargeted MA under demoiréing methods.

| Untargeted MA | Inception-V3 | Demoiréing method | | | |
|---|---|---|---|---|---|
| | | MopNet | MDDM | FHDe2Net | AMNet |
| $\epsilon = 4$ | 1.000 | 0.989 | 0.997 | 1.000 | 0.974 |
| $\epsilon = 8$ | 1.000 | 0.994 | 1.000 | 1.000 | 0.989 |

We find that moiré pattern is actually annoying and intractable to be greatly removed. Even moiré pattern is partly mitigated under such demoiréing methods in perception, the image still remains

adversarial to the victim CNN model (as shown in Table 5, the success rate is still high under demoiréing methods). In this view, moiré pattern not only causes visual impediment but can really be a threat to the safety of CNN, which should be aware and more exploration should be devoted to remove or weaken its potential threat.

## 4.4 Transferability analysis

In this part, we study the transferability of the proposed Moiré Attack. We choose Inception-V3 [37] as the source model to attack and Resnet34 [19], Resnet50 [19], Densenet121 [20], VGG16 [35], and VGG19 [35] as the remote target models to test the generated adversarial examples. The noise budget $\epsilon = 4$. All the attacks are untargeted attack. Table 6 shows the proposed Moiré Attack (MA) has high transferability across different DNNs (the success rates are all about 95%), while the baselines behave poorly in the transferability.

Table 6: The attack success rate of different models. The second left column is the source model Inception-V3 and the right 5 columns are the remote target models.

| Attacks | Inception-V3 | Resnet50 | Resnet34 | Densenet121 | VGG16 | VGG-19 |
|---------|--------------|----------|----------|-------------|-------|--------|
| MA | **1.000** | **0.967** | **0.927** | **0.959** | **0.982** | **0.973** |
| BIM | 0.989 | 0.176 | 0.202 | 0.173 | 0.233 | 0.202 |
| PGD | 0.990 | 0.166 | 0.202 | 0.167 | 0.224 | 0.201 |
| CW | 0.762 | 0.116 | 0.133 | 0.121 | 0.170 | 0.146 |
| MI-FGSM | 0.991 | 0.254 | 0.294 | 0.247 | 0.312 | 0.298 |

## 4.5 Ablation Study

**Budget-free Moiré Attack.** To explore the relationship between the sensor noise and the final generated moiré pattern in perception, we set the sensor noise budget $\epsilon$ from 4 to 32. Te details of the generated adversarial examples are shown in Figure 6.

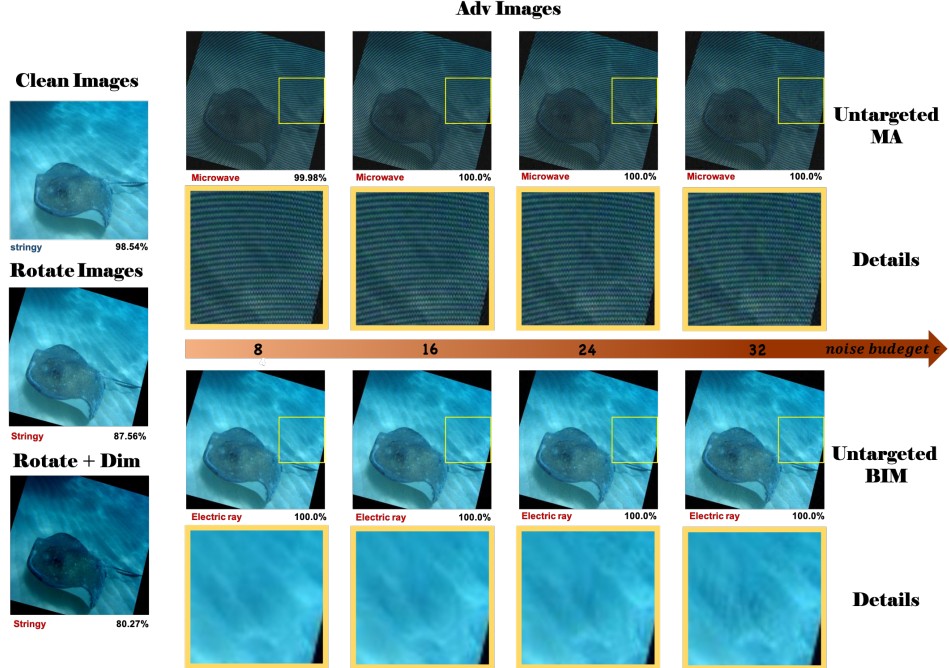

Figure 6: Comparison of the details of the adversarial examples generated by Moiré Attack (MA) and BIM.

We can see that with the increase of the added sensor noise budget $\epsilon$, the details of moiré in the adversarial example hardly change, which means there is no obvious change perceptible to human eyes. While for BIM, when the perturbation is up to a threshold (e.g. 32), the noise can be easily noted by human eyes. In this aspect, our proposed Moiré Attack is a budget-free attack.

**Factors affecting the synthesis moiré pattern.** In this part, we further explore different factors that will influence the generated moiré pattern including the shooting rotation angles $\gamma$ and the size of the LCD monitor (size scale factor $\alpha$). The sensor noise budget is $\epsilon = 4$. The results are shown in Figure 7. For rotation angles, the results show that moiré patterns with different rotation angles generated by Moiré Attack are different. Another factor influencing the moiré pattern is the size scale degree when displaying an image on the monitor. We can see that when the size of the monitor increases, the generated moiré pattern becomes more obvious and the distortion is more evident. In addition, we also discuss the position of the introduced fabricated noise' position to MA in Appendix E.

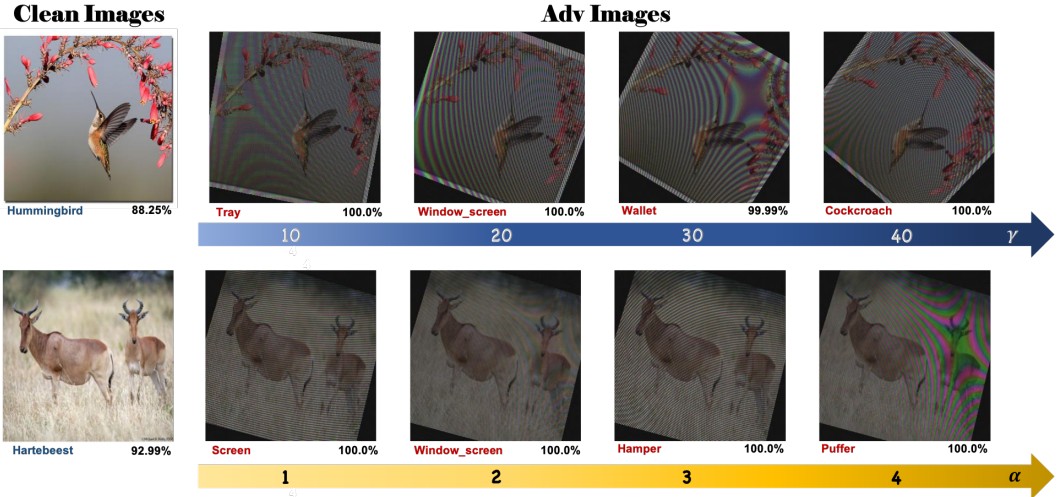

Figure 7: Factors that influence the generated moiré pattern. The top row shows the different moiré pattern with different rotation angles $\gamma$ and the bottom row shows moiré pattern under different size scale factor $\alpha$.

## 5    Conclusion and Future work

In this paper, we revealed the threats of DNNs from a new perspective. We utilized the physical property of the camera when shooting, i.e., the moiré effect, and demonstrated its potential security risk for DNNs. Then we proposed Moiré Attack by mimicking the shooting process of a camera. Extensive experiments have been conducted to present great properties of the proposed Moiré Attack, e.g., high attack success rate, high transferability across different models, and high robustness under various image transformation methods. However, the sensor noise is uncontrollable in a sense and it may be difficult when delivering the targeted MA in practice. Our work is just a small step to open up a new possibility to craft attacks against DNNs. We will continue to explore more general moiré effect generation methods.

## Acknowledgments and Disclosure of Funding

Yisen Wang is partially supported by the National Natural Science Foundation of China under Grant 62006153, and Project 2020BD006 supported by PKU-Baidu Fund. We would like to thank Hanyue Lou at Peking University in the discussion at moiré phenomenon in daily life.

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
