## A  The algorithm for Moiré Attack (MA)

---

**Algorithm 1** Moiré Attack

---

**Input:** clean image $x$; targeted label $x^*$; ground truth label $y_{truth}$; a targeted class $y_{adv}$; a DNN classifier $f$ with cross-entropy loss function $\mathcal{L}_{adv}$; differential transformation of camera processing $g_1$ (step 1 to step 5 in section 3.3) and $g_2$ (step 6 in section 3.3), sensor noise $\delta$, sensor noise budget $\epsilon$; iteration time $K$; camera rotation angle $\gamma$, monitor displaying size scale factor $\alpha$;

**Output:** Adversarial example $x^*$

1: $\delta = 0$
2: $\epsilon\prime = \frac{\epsilon}{K}$
3: Mimic the shooting process of camera on the LCD monitor and obtain the image data $g_1(x, \delta, \gamma, \alpha)$
4: $x_0^* = g_1(x, \delta, \gamma, \alpha)$
5: **for** $k = 0$ to $K - 1$ **do**
6:    Input $x_k^*$ to $f$ and obtain $\nabla_\delta \mathcal{L}_{adv}(x_k^*, y_{adv})$ for targeted MA or $\nabla_\delta \mathcal{L}_{adv}(x_k^*, y_{truth})$ for untargeted MA
7:    Update the sensor noise $\delta$: $\delta = clip_{\epsilon,\delta} \{\delta - \epsilon\prime * sign(\nabla_\delta \mathcal{L}_{adv})\}$
8:    Update $x^*$: $x_{k+1}^* = clip_{([0,255],x)} \{x_k^* + \delta\}$
9: **end for**
10: Obtain adversarial example $x^*$ by using $g_2$ to recover $x_K^*$ to RGB format: $x_K^* = g_2(x_K^*)$
11: **return** $x^* = x_K^*$

---

## B  The comparison of Moiré Attack with other physical attacks

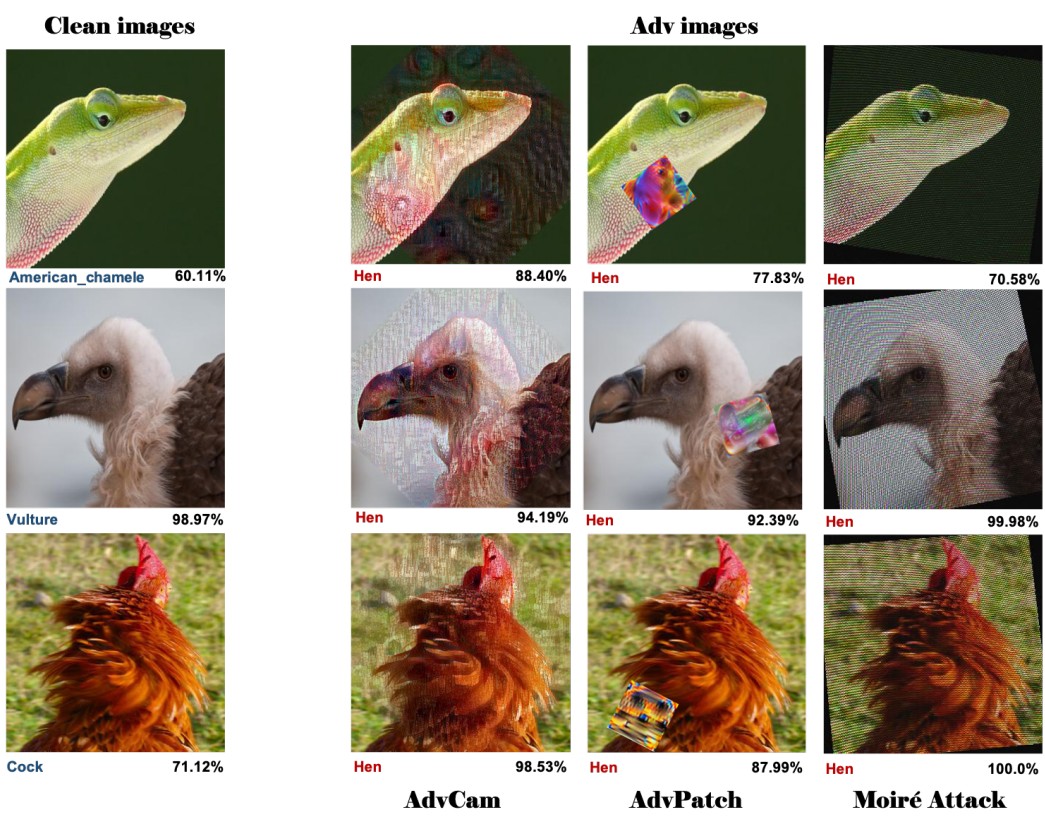

Figure 1: The comparison of targeted MA with other successful physical-world attacks in human perception.

With the purpose to disguise the attack into an unavoidable common phenomenon - moiré pattern, MA can be regarded as a combination of physical attack and digital attack. It not only has high success rate, but seems more natural compared with common physical attacks in the perspective of the probability to catch people's attention. For visualization, we adopt several physical attack methods such as AdvPatch [1] and AdvCam [2] for comparison. The generated adversarial examples are included in Figure 1 , we can see that although all of them are distorted, MA seems more natural.

## C    The perception of Moiré Attack

As mentioned in Section 3.1, moiré pattern can be a potential threat to DNNs. However, it hardly arouses humans' attention when it is inevitably generated through shooting on the LCD screens, where humans will consider it as a normal and unavoidable phenomenon, even if their eyes do capture the evident distortion. In this view, the distortion of morié pattern is reasonable and rational for human eyes and perception. It is also the exact precondition and motivation of the proposed MA. In our practical attack process, we fabricate morié pattern by mimicking the shooting process. It should guarantee that the simulation moiré pattern is very similar to the real one that at least human eyes are unable to distinguish them.

For the method to mimic the shooting process, the challenge on image demoiréing [5, 4] uses an official dataset called LCDMoiré [5] for benchmarking example-based image demoiréing. We strictly follow the same procedure in our simulation of moiré pattern. Our generated images are also consistent with the ones in the literature [5, 3]. Some images in LCDMoiré and their clean counterpartas are shown in Figure 2 (a) (b). We find that the synthesis images are darker and distorted to some degree compared with the original ones. It is also consistent with the analysis in the step 2 in Section 3.3 that the unavoidable real process of resampling the pixels into subpixels indirectly causes that distortion.

To validate the rationality of resulting darkness of the generated moiré pattern, we also use iPhone 12 rear camera to take pictures on the AOCU27P1U 4K 3840*2160dpi monitor amd MacBook Air 2020 2560*1600dpi monitor. The comparisons are shown in Figure 2 (c) and (d). We can see that the images shot on LCD are darker in general, especially when the brightness of the monitor is higher.

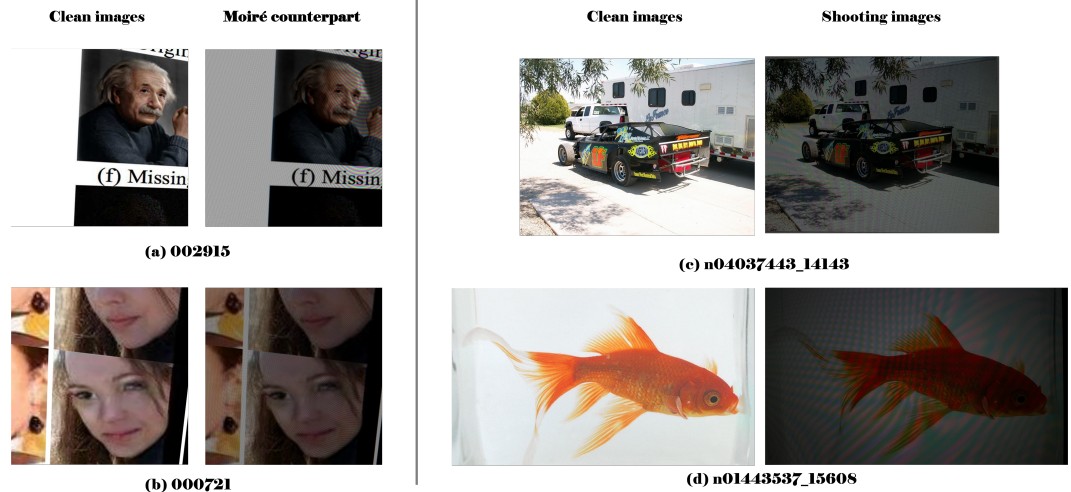

Figure 2: Images with moiré pattern. (a) and (b) are selected from MoiréDataset. In (c) and (d), clean images are from ImageNet Val set. Shooting camera is iPhone 12 rear camera, the monitor of (c) is AOCU27P1U 4K 3840*2160dpi, the monitor of (d) is MacBook Air 2020.

# D    The visualization of Morié Attack under demoiréing method

The visualization of MA under the state-of-the-art demoiréing methods is shown in Figure 3.

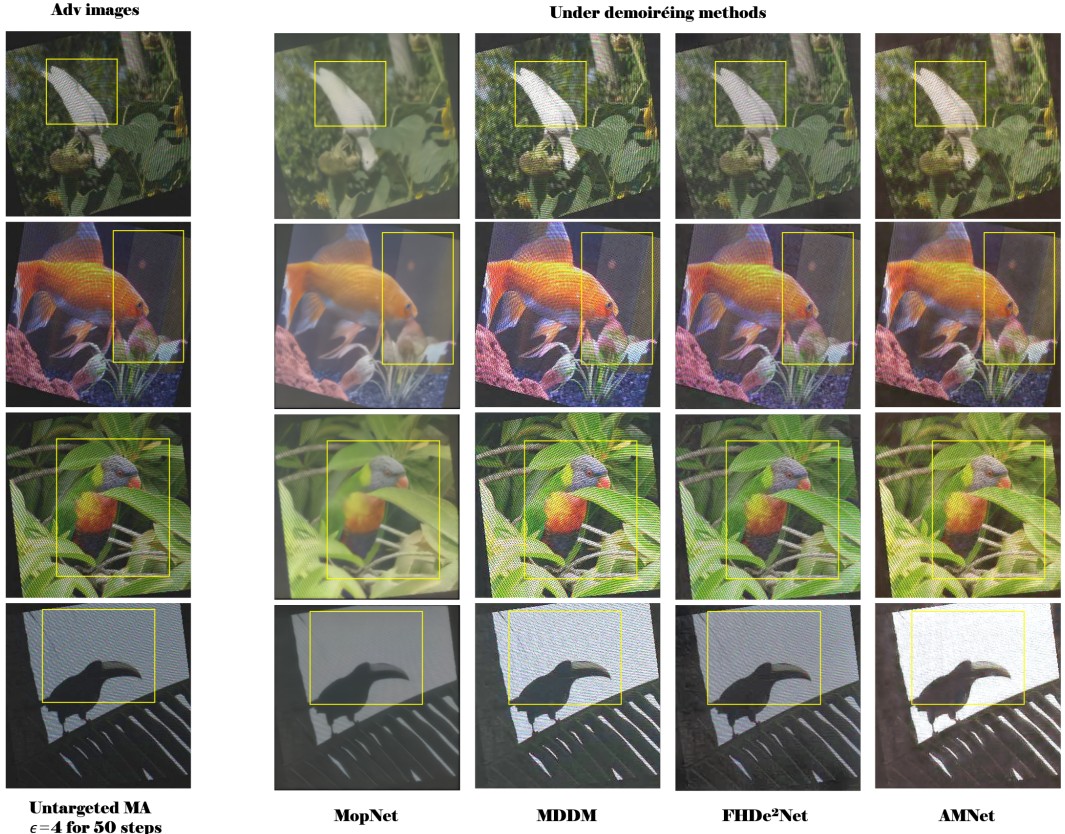

Figure 3: The qualitative results of untargeted MA under demoiréing methods.

# E    The position of the added noise

In our pipeline of MA, we add fabricated noise in step 5. Step 4 mimics how the sensor of the camera reads the raw image data, which in fact transfers the three-dimensional RGB data to a two-dimensional RGGB pattern. Step 6 is the inverse process of step 4 which recovers the data to a normal image. Between them, camera sensor inevitably brings some noise in step 5, which is exactly our motivation to propose MA, i.e., the sensor noise is unavoidable that can be utilized by the attacker.

In our ablation experiment, we also perform the attack in step 6, which in fact a simple combination of the shooting process and an attack method like PGD. We compare untargeted MA to a simple baseline with noise added in step 6 (Moiré + BIM). The success rate before and after JPEG compression shown in Table 1. We can see that in Moiré + BIM, the noise is easier to be removed by compressing the adversarial example. In this aspect, the proposed MA is more robust under image transformations.

Table 1: The success rate of untargeted MA and Moiré + BIM under JPEG compression. The noise budget $\epsilon$ is 4.

| Attack | Before JPEG compression | qf=20 | qf=40 |
|---|---|---|---|
| untargeted MA | **1.000** | **0.938** | **0.949** |
| Moire + BIM | 0.998 | 0.857 | 0.902 |