# OpenReview forum: "Morié Attack (MA): A New Potential Risk of Screen Photos"
_NeurIPS.cc/2021/Conference — NeurIPS 2021 Poster_

### Official Review · Reviewer_iszk · 2021-07-14

**Rating:** 8
**Confidence:** 4

**Summary:**

The paper found a very interesting phenomenon in current digital devices, called morie effect, that can be used for attack. They proposed a Morie Attack (MA) that generates the physical-world morie pattern adding to the images by mimicking the shooting process of digital devices. MA is with great stealthiness that hardly attracts the awareness of humans. Extensive experiments show that MA can attack DNN with a high success rate and high transferability rate.

**Limitations And Societal Impact:**

Pls refer to weaknesses.

**Main Review:**

Strength:
(i) The morie effect maybe heard by many people, but it is very interesting to see that it can be used as an attack method. There is a fact we maybe ignored before that the final perceived images by human-vision and computer-vision systems could be very different in some cases, for example shooting on digital monitors due to the different physical mechanisms. Thus, the exploration in this paper is novel and is clearly of interest for the adversarial community.
(ii) The proposed morie attack is simple but effective. The authors mimic the shooting process of the camera sensors, which gives the attackers more freedom and controllability to tamper with DNNs. And the generated morie effects seem real and hardly distinguishable for human beings.
(iii) The experiments seem to be completed and comprehensive. The authors have conducted the targeted and untargeted evaluations, and also checked the effectiveness of morie attack under various transformations. Detailed ablation studies convince the effectiveness of morie attack.

Weakness:
(i) Is the morie attack sensitive to the angle of shooting devices?

(ii) How the perturbation is added into the generation of morie process? It is better if the authors provide an algorithm to show the process of morie attack.

(iii) Why MA is so effective under median smoothing in Table 3?



**Time Spent Reviewing:**

6

---

> ### Author Response · Authors · 2021-08-10
> **Response to Reviewer iszk**
>
> Thanks for your insightful comments and useful suggestions. Please find our responses below to your questions.
> ***
> Q1: Is moiré attack sensitive to the shooting angles?
>
> A1: Yes, shooting angle directly affects the generated moiré pattern. In the second part of our ablation study, we explore the factors that may result in visible difference on moiré pattern. As Figure 7 shows, with different camera’s rotation angle $\gamma$, the generated adversarial example presents visibly different moiré pattern.
> ***
> Q2: It is better if the authors provide an algorithm to show the process of moiré attack (MA)
>
> A2: A good idea. We will add the algorithm in the revised version as follows:
>
>
> ***
> **Algorithm 1**  Moiré Attack
> ***
> **Input:** clean image $x$; ground truth label $y_{t}$; a targeted class $y_{a}$;  a DNN classifier $f$ with cross-entropy loss function $\mathcal{L}_{a}$; differential transformation of camera processing $g_1$ (step 1 $\sim$ step 5 in section 3.3) and $g_2$ (step 6 in section 3.3); sensor noise $\delta$; sensor noise budget $\epsilon$; iteration time $K$; camera rotation angle $\gamma$, monitor displaying size scale factor $\alpha$;
>
> **Output:** Adversarial example $x^*$
>
> 1: &nbsp; $\delta$ = 0
>
> 2: &nbsp; $\epsilon \prime = \frac{\epsilon}{K}$
>
> 3: &nbsp; Mimic the shooting process of camera on LCD monitor and obtain the image data $g_1(x, \delta, \gamma, \alpha)$
>
> 4: &nbsp; $x_0^* = g_1(x, \delta, \gamma, \alpha)$
>
> 5: &nbsp; **for** $k=0$ to $K-1$ **do**:
>
> 6: &nbsp; &emsp; Input $x^*_k$ to $f$ and obtain  $\nabla_\delta \mathcal{L}_a (x^*_k, y_a)$ for targeted MA or $\nabla_\delta \mathcal{L}_a(x^*_k, -y_t)$ for untargeted MA
>
> 7: &nbsp; &emsp; Update the sensor noise $\delta$: $\delta$ = $clip_{\epsilon, \delta}$ { ${ \delta - \epsilon \prime * sign(\nabla_{\delta} \mathcal{L}_{a} \} $ ) }
>
> 8: &nbsp; &emsp; Update $x^*$: $x^*_{k+1} = clip_{ ( [0, 255], x ) } $ { $x^*_k + \delta$ }
>
> 9: &nbsp; **end for**
>
> 10: &nbsp; Obtain adversarial example $x^*$ by using $g_2$ to recover $x^*_K$ to RGB format: $x^*_K = g_2(x^*_K)$
>
> 11: &nbsp; **return** $x^*=x^*_K$
> ***
> ***
> Q3: Why MA is so effective under median smoothing in Table 3?
>
> A3: Median smoothing (also known as median blur or median filter) is a kind of transformations to mitigate the noise in an image [1]. It runs a sliding window over each pixel in an image, where the center pixel is replaced by the median value of its neighboring pixels within the window. The process spreads pixel values across nearby pixels and makes adjacent pixels more similar. Median smoothing is very effective to remove the noise with periodic patterns (lines evenly or similarly spaced) for it can average the pixel values between the lines, therefore, blurring the difference between the periodic noise and the content. As we analyze, the moiré effect is a large-scale interference phenomenon and the moiré pattern appears as periodic lines which can be partly removed by median smoothing. This attributes to the relatively low success rate of MA in Table 3 compared with other attack methods. While as for BIM, PGD, and MI-FGSM, the adversarial noise is introduced intentionally instead of randomly, the deliberately fabricated perturbation usually bestows the adversarial example some new hidden features that mislead the CNN to make a wrong prediction. The hidden feature is inter-correlative in the high dimension, thus it can be hardly removed by median smoothing (a pixel-range transformation), that is why BIM, PGD, and MI-FGSM perform a little bit better under median smoothing.
> ***
> [1]Huang, Thomas, G. J. T. G. Y. Yang, and Greory Tang. "A fast two-dimensional median filtering algorithm." IEEE transactions on acoustics, speech, and signal processing 27.1 (1979): 13-18.

---

> > ### Comment · Reviewer_iszk · 2021-08-16
> > **After rebuttal**
> >
> > Thanks to the authors' rebuttal. My major concern has been solved.

---

### Official Review · Reviewer_FN6p · 2021-07-14

**Rating:** 6
**Confidence:** 5

**Summary:**

This paper discovers a special phenomenon, the Moire effect, that can be used for adversarial attack. A Moire Attack method is proposed by mimicking the shooting process of the camera sensors by optimizing the sensor noise. Empirically, this method has a high success rate and transferability across different victim models.

**Limitations And Societal Impact:**

Some alternative attacks based on the Moire effect can be explored, e.g. optimization on image instead of on the sensor noise, which can eliminate the concerns.

**Main Review:**

Originality:
This paper discovers a novel way to construct an adversarial attack.

Quality:
I have several technical concerns.
1. Should the 'Morié effect' be 'Moiré effect' in the title?
2. I think the perturbation budget $\epsilon$ should be a constraint on x rather than on the sensor noise $\delta$, as the same budget on $\delta$ will result in much more significant changes on the image x perceived by DNNs.
3. Can the authors show the images after demoire algorithms? Will there be some special patterns when the moire patterns are removed?
4. It seems the proposed method depends on the manipulation of the optical image capture pipeline, which I believe is in practice embedded in a physical camera and cannot be altered by hand. This makes the proposed method impractical as, for my understanding, no one can actually control the perturbation on the sensor noise in a camera. It is much more convincing to show successful attacks on physical cameras.
5. The last point also raises another one: it seems the method is device-specific, e.g. it can only impact a single device even the attack method can be deployed on a physical camera device. On contrary, most existing adversarial attacks generate perturbations on raw images, leading to general attacks that can affect many devices.

Clarity:
This paper is clear and easy to follow.

Significance:
This is an interesting direction of research. Maybe there are some alternative ways to construct the Moire Attack that can avoid the problems I am concerned with. Therefore I believe this is a fairly significant paper.

**Time Spent Reviewing:**

3

---

> ### Author Response · Authors · 2021-08-10
> **Response to Reviewer FN6p**
>
> Thanks for your insightful comments and useful suggestions. Please find our responses below to your questions.
> ***
> Q1: Should ‘Morie effect' be Moiré effect’ in the title?
>
> A1: Yes, thanks for pointing it out. We will fix it in the revision.
> ***
> Q2: The perturbation budget $\epsilon$ should constraint on $x$ rather than sensor noise $\delta$.
>
> A2: For common white-box attacks, it is necessary to constrain the final output image $x^*$ for its purpose to make the adversarial noise small enough to be imperceptible for human eyes. However, moiré attack itself is and should be a visible attack that does not need such a constraint (we also demonstrate its budget-free quality in the first part of our ablation study): the human will deem it reasonable even the adversarial examples with moiré pattern seem to be distorted compared with the original clean image, because such a distortion is rational as it is a real-world phenomenon. In this aspect, MA can be considered as a physical attack. If you use your phone to take a phone of an image on your screen and use it to subtract the clean one, you will find the difference between them is very large because of the shooting angle, the background light and the moiré pattern, while you may not think the obtained image is questionable because all of such distortions are reasonable and unavoidable.
> ***
> Q3: Can the authors show the images after demoiréing algorithms? Will there be some special patterns when the moiré patterns are removed?
>
> A3: To explore whether MA will be affected by the extant demoiréing methods, we add extra experiments here. We search for the *challenge of demoiréing* [1] [2] in recent years and choose four SOTA methods MopNet [3], MDDN [4], FHde2Net [5], AMNet [6] to test MA (the settings are the same as the ones in their papers). We show the success rate of MA under demoiréing methods in Table 5 and the visualization of demoiréing results in Figure 9 (untargeted MA, $\epsilon = 4$) (https://www.dropbox.com/s/xr7fgbvv1ntumyn/Figure.png?dl=0). We find that moiré pattern is actually annoying and intractable to be greatly removed. Even moiré pattern is partly mitigated under such demoiréing methods in perception, the image is very likely to still remain adversarial to the victim CNN model (as shown in Table 5, the success rate is still high under demoiréing methods). In this view, moiré pattern not only causes visual impediment but can really be a threat to the safety of CNN, which should be aware and more exploration should be devoted to remove or weaken its potential threat. We would like to conduct more research on this direction in the future.
>
> **Table 5: The success rate of untargeted MA under demoiréing methods.**
>
> | Demoiréing method | MopNet | MDDM | FHDe2Net | AMNet |
> |:--------------------------:|:-----------:|:---------:|:--------------:|:----------:|
> | $\epsilon = 4$          | 0.989     | 0.997   | 1.000         | 0.974   |
> | $\epsilon = 8$          | 0.994     | 1.000   | 1.000         | 0.989   |
>
> ***
> Q4: The proposed MA is impractical because we cannot control the optical image process by our hand.
>
> A4: In Section 3.1, through experiments, we explore and find the unavoidable moiré pattern caused by shooting on the LCD monitor will be a threat and can mislead CNNs to make wrong predictions. Such a finding is encouraging and motivates us to propose MA. In fact, the moiré pattern in the image itself can mislead the CNNs, while such misleading is untargeted. We agree with you that the sensor noise is uncontrollable and it may be difficult when delivering the targeted MA in practice. While we want to argue that our work is just a small step in the way to study moiré effect and adversarial examples. We will definitely try and continue to explore more general moiré effect generation methods instead of sensor noise in the future.
> ***
> Q5: MA is device-special that can only impact a single device.
>
> A5: One point should be clarified is that the output image generated by MA is the adversarial example itself, which has high transferability and remains adversarial to different CNNs no matter which device is involved in the attack. Just like we have stated in Lines 134 - 141, we use different devices and LCD players to explore the threat of moiré pattern. Figure 3 shows that moiré pattern itself is enough to mislead the CNNs as an untargeted attack no matter which device we use.
> ***
> [1] Yuan, Shanxin, et al. "Ntire 2020 challenge on image demoiréing: Methods and results." Proceedings of the IEEE/CVF Conference on Computer Vision and Pattern Recognition Workshops. 2020.
>
> [2] Yuan, Shanxin, et al. "Aim 2019 challenge on image demoiréing: Dataset and study." 2019 IEEE/CVF International Conference on Computer Vision Workshop (ICCVW). IEEE, 2019.
>
> [3] He, Bin, et al. "Mop moiré patterns using mopnet." Proceedings of the IEEE/CVF International Conference on Computer Vision. 2019.
>
> [4] Cheng, Xi, Zhenyong Fu, and Jian Yang. "Multi-scale dynamic feature encoding network for image demoiréing." 2019 IEEE/CVF International Conference on Computer Vision Workshop (ICCVW). IEEE, 2019.
>
> [5] He, Bin, et al. "FHDe 2 Net: Full High Definition Demoiréing Network." Computer Vision–ECCV 2020: 16th European Conference, Glasgow, UK, August 23–28, 2020, Proceedings, Part XXII 16. Springer International Publishing, 2020.
>
> [6] Yue, Huanjing, et al. "Recaptured screen image demoiréing." IEEE Transactions on Circuits and Systems for Video Technology 31.1 (2020): 49-60.

---

> > ### Comment · Reviewer_FN6p · 2021-08-16
> > **After rebuttal comment**
> >
> > Thanks for the author's response. My main concerns are resolved, and I will keep my rating positive.

---

### Official Review · Reviewer_LYp3 · 2021-07-16

**Rating:** 5
**Confidence:** 3

**Summary:**

A new white-box attack to deep neural network is proposed. The authors observe that a large number of digital images are affected by the so called moiré effect or moirè pattern. When images are displayed on a digital monitor they are obviously sampled, then they are sampled again when acquired by a new device. The display and acquisition sampling grids are usually non-aligned, which gives rise to visible distortion patterns of varied geometry in the acquired image, known as moiré patterns. The proposed attack takes a generic image, introduces a synthetic moiré pattern on it, by replicating its generation process, and then attacks it by conventional methods. Experiments support the effectiveness of the proposed method.


**Limitations And Societal Impact:**

The authors did not describe limitations and potential negative societal impacts of their work. For what concern limitations it would be important to show what happens when it is applied a method to detect and even remove the moirè pattern.

**Main Review:**

A first observation is on the language, which should be definitely improved. There are many errors and typos, to begin with moiré which is consistently (108 times) spelled morié. The organization is also questionable, with too much space devoted to relatively well-known facts (the moiré effect) and too little to experimental validation. Even the presentation needs to be improved. Apart from this, my main criticism is methodological. As far as I understand, in fact, no new attack is really proposed but rather a means to disguise an attack. The original image is not attacked: only a strongly distorted version of it is. As the authors observe, moiré patterns occur frequently in images, so this type of added distortion may remain unnoticed. However, many images are also highly noisy, under or overexposed, almost all of them are compressed, so one could use any of these facts or a combination of them to disguise attacks. Accordingly, the proposed cannot be fairly compared with methods that work on the original image, but rather with other "attack disguising" methods. Moreover, as the authors themselves acknowledge, methods to reduce or remove the moiré patterns exist and should be taken into account, e.g. [*].

Concerning the implementation of the attack, the authors remark that "step 2 causes the final images to be darker" and in fact all visual examples show very dark attacked images. I find this quite weird, considering that attacked images should look natural, not to raise suspects (this is in the spirit of mimicking moiré patterns). These images could be easily singled out and subject to a deeper scrutiny just based on their mean value. In addition, it is very difficult to asses their quality by visual inspection. Also, it appears that after the backpropagation-based attack (step 5) the image is subject to further processing (step 6) which could weaken it. I do not understand why the attack is not performed as the last step.

Experiments, though encouraging, are very limited and do not consider several datasets usually adopted in the field. Finally, the subsection ablation study does not present any ablation, but other types of analyses.

Overall, I believe that this paper needs substantial work to make it acceptable for possible publication.

References
- [*] He et al. Mop Moiré Patterns Using MopNet, ICCV 2019.

--------------------------------------------------------------------------------------------------

UPDATES

First of all, I want to thank the authors for answering my questions.

Some of my concerns have been solved, in particular authors clarified the type of attack,
they analyzed methods to remove the Moiré patterns and ensured that they will improve the presentation.
They also added a comparison with a simple baseline to show that their proposal works better.
I still have a concern about the visible and strong distortion introduced with this attack since I believe that
it is not reasonable and indeed it would raise human attention. In this respect in my opinion comparisons with state-of-the-art are not fair.
However, given that some major issues have been addressed, I increased my score.

**Time Spent Reviewing:**

4

---

> ### Author Response · Authors · 2021-08-10
> **Response to Reviewer LYp3**
>
> Thanks for your valuable comments. We try our best to address your concerns.
> ***
> Q1: Typos and language errors.
>
> A1: Thanks for pointing them out. We will fix them in the revision.
> ***
> Q2: Questionable organization with too much space devoted to moiré effect and too little to experiments.
>
> A2: In Section 3.1, we only use half of a page space to simply introduce the moiré effect. The rest of Section 3.1 is mainly to investigate the threat of moiré pattern to CNN, which is novel and also exactly our motivation to propose moiré attack (MA). As for the experiments, we first evaluate the effectiveness of MA by testing the success rate under targeted and untargeted MA. Then, we analyze the robustness of MA under JPEG compression and different transformation methods. We also test its adversarial transferability through different models. In the ablation study, we further explore the factors $\gamma$, $\alpha$, and $\epsilon$ that affect the generated moiré patterns. We believe the experiments are relatively comprehensive to evaluate a new proposed attack.
> ***
> Q3: No new attack is proposed but rather a method to disguise the attack.
>
> A3: We respectfully disagree. One point that should be clarified is that the output image generated by MA is an adversarial example itself, which mimics the real processing of camera shooting on LCD, so it should be perceived and compared with the photos taken by a camera in front of LCD rather than the original clear one. In this view, the distortion on it is reasonable and rational for human eyes and perception. All these are preconditions and motivations of our proposed MA, an attack method to generate adversarial examples by utilizing an unavoidable physical phenomenon of moiré effect. Note that the methodology of MA is not generating moiré effect first and then adding adversarial noise on it (Please refer to the following Q7\&A7).
> ***
> Q4: It is unfair to compare MA with the methods which deliver the attack on the original images.
>
> A4: As mentioned in A3, the adversarial examples of MA should be perceived and compared with the photos taken by a camera on LCD rather than the original clean digital images. In fact, MA can be regarded as a combination of physical attack and digital attack. MA not only has high success rate, but also seems more natural compared with other physical attacks. We further adopt several physical attack methods such as AdvPatch [3] and AdvCam [4] for comparison. The generated adversarial examples are included in Figure 8 in <https://www.dropbox.com/s/xr7fgbvv1ntumyn/Figure.png?dl=0>, we can see that all of them are distorted while MA seems more natural.
> ***
> Q5: Methods to reduce or remove the moiré patterns exist and should be taken into account, e.g. [5].
>
> A5: To explore whether MA will be affected by the extant demoiréing methods, we add extra experiments here. We search for the *challenge of demoiréing* [1] [2] in recent years and choose four SOTA methods MopNet [5], MDDN [6], FHde2Net [7], AMNet [8] to test MA (the settings are the same as the ones in their papers). We show the success rate of MA under demoiréing methods in Table 5 and the visualization of demoiréing results in Figure 9 (untargeted MA, $\epsilon = 4$) (https://www.dropbox.com/s/xr7fgbvv1ntumyn/Figure.png?dl=0). We find that moiré pattern is actually annoying and intractable to be greatly removed. Even moiré pattern is partly mitigated under such demoiréing methods in perception, the image is very likely to still remain adversarial to the victim CNN model (as shown in Table 5, the success rate is still high under demoiréing methods). In this view, moiré pattern not only causes visual impediment but can really be a threat to the safety of CNN, which should be aware and more exploration should be devoted to remove or weaken its potential threat. We would like to conduct more research on this direction in the future.
>
> **Table 5: The success rate of untargeted MA under demoiréing methods.**
>
> | Demoiréing method | MopNet | MDDM | FHDe2Net | AMNet |
> |:--------------------------:|:-----------:|:---------:|:--------------:|:----------:|
> | $\epsilon = 4$          | 0.989     | 0.997   | 1.000         | 0.974   |
> | $\epsilon = 8$          | 0.994     | 1.000   | 1.000         | 0.989   |
>
> ***
> Q6: The generated adversarial examples are darker, weird and can be singled out based on their mean value.
>
> A6: We agree with you that it may be a little bit weird when first observing the generated examples with moiré patterns, but the phenomenon does exist in the real process of shooting images on the LCD monitor in practice. In fact, the feeling of abnormality is caused by the comparison with the clean original image, while we are taking images on LCD monitors. Maybe you can try now and use your mobile phone to take a photo of the screen in front of you. You will find that the obtained image will be distorted and darker compared with the original one. It is also consistent with the analysis in lines 185 - 188 that the unavoidable real process of resampling the pixels into subpixels indirectly causes that distortion.
>
> As for whether it can be easily singled out based on their mean value, we further add experiments to check this. We set the mean value of the final adversarial examples the same as the clean images by subtracting their difference. We set the sensor noise budget $\epsilon$ = 2, 4, 6, 8 and the step = 50. The attack success rate is shown in Table 6 , we can see MA is relatively robust to such a mean value normalized processing.
>
> **Table 6: The success rate of untargeted MA after mean value normalizing.**
>
> | &emsp;&emsp;&emsp;&emsp;&emsp;Attack                      | &emsp;&emsp;&emsp;Untargeted MA|
> |:--------------------------:|:------------------:|
> | Noise budget $\epsilon$              | 2 &emsp;&emsp;&emsp; 4 &emsp;&emsp;&emsp; 6 &emsp;&emsp;&emsp; 8     |
> | Succ (mean value normalized)    | 0.960 &emsp; 0.987&emsp;  0.994 &emsp; 0.998   |
>
> ***
> Q7: Why the attack is not performed in the last step?
>
> A7: First, step 4 mimics how the sensor of the camera reads the raw image data, which in fact transfers the three-dimensional RGB data to a two-dimensional RGGB pattern. Step 6 is the inversive process of step 4 which recovers the data to a normal image. Between them, camera sensor inevitably brings some noise in step 5, which is exactly our motivation to propose MA, i.e., the sensor noise is unavoidable that can be utilized by the attacker. Thus, we perform attack in the step 5.
>
> If we perform the attack in step 6, it just simply combines the shooting process and an attack method like PGD, which is very trivial. Moreover, if the noise is directly added after step 6, it is easy to be separated and removed by directly compressing it after step 6, whereas the noise in step 5 is difficult to remove because it is also be demosaicing in step 6 (You can also refer Q5\&A5 to see the results of demoiréing).
> ***
> Q8: The experiment does not consider several datasets in this field.
>
> A8: We randomly selected 5000 images in ImageNet testdev which have been correspondingly tested to be correctly classified by the victim CNN. ImageNet is a large and comprehensive dataset, and most of the universal attack methods (that means the attack does not aim at a special field like face recognition) usually adopt ImageNet to do the experiments. For MA, we evaluate its effectiveness by testing the success rate under targeted and untargeted MA. Then, we analyze the robustness of MA under JPEG compression and different transformation methods. We also test its adversarial transferability through different models. In the ablation study, we further explore the factors $\gamma$, $\alpha$, and $\epsilon$ that affect the generated moiré patterns. We believe the experiments are relatively comprehensive to evaluate a new proposed attack. Finally, we are willing to test MA on more datasets in the revision.
> ***
> Q9: The ablation study does not present any ablation, but other types of analyses.
>
> A9: We respectfully disagree. In our ablation study, we explore different factors that may affect the generated moiré pattern. We correspondingly set the parameter $\gamma$, $\alpha$, $\epsilon$ (the definitions of them refer to Section 3.3) to different values in our attack process and compare the attack results of them.
> ***
> [1] Yuan, Shanxin, et al. "Ntire 2020 challenge on image demoiréing: Methods and results." Proceedings of the IEEE/CVF Conference on Computer Vision and Pattern Recognition Workshops. 2020.
>
> [2] Yuan, Shanxin, et al. "Aim 2019 challenge on image demoiréing: Dataset and study." 2019 IEEE/CVF International Conference on Computer Vision Workshop (ICCVW). IEEE, 2019.
>
> [3] Brown, Tom B., et al. "Adversarial patch." arXiv preprint arXiv:1712.09665 (2017).
>
> [4] Duan, Ranjie, et al. "Adversarial camouflage: Hiding physical-world attacks with natural styles." Proceedings of the IEEE/CVF conference on computer vision and pattern recognition. 2020.
>
> [5] He, Bin, et al. "Mop moiré patterns using mopnet." Proceedings of the IEEE/CVF International Conference on Computer Vision. 2019.
>
> [6] Cheng, Xi, Zhenyong Fu, and Jian Yang. "Multi-scale dynamic feature encoding network for image demoiréing." 2019 IEEE/CVF International Conference on Computer Vision Workshop (ICCVW). IEEE, 2019.
>
> [7] He, Bin, et al. "FHDe 2 Net: Full High Definition Demoiréing Network." Computer Vision–ECCV 2020: 16th European Conference, Glasgow, UK, August 23–28, 2020, Proceedings, Part XXII 16. Springer International Publishing, 2020.
>
> [8] Yue, Huanjing, et al. "Recaptured screen image demoiréing." IEEE Transactions on Circuits and Systems for Video Technology 31.1 (2020): 49-60.

---

> > ### Comment · Reviewer_LYp3 · 2021-08-24
> > **Need more clarifications**
> >
> > Dear authors,
> >
> > I want to thank you for answering my questions.
> > In particular, it is more clear to me now that this is an attack and not just a way to hide adversarial perturbations (I was confused by the sentence ''the morié pattern can be a perfect camouflage of the adversarial perturbations'', lines 60-61). Thanks also for analyzing methods to remove the Moiré patterns, the results you obtained are very interesting.
> >
> > However, I still have some concerns about the fact that the introduced attack causes a very visible distortion on the image. Now I tried your experiment and used my mobile phone to take a photo of my screen, but did not obtain such a distorted and dark image as those presented in this paper. I also checked the paper on the ''NTIRE 2020 Challenge on Image Demoireing'' and again I was not able to find so deeply distorted images (see Fig.1, top row).
> >
> > In addition, in my opinion comparisons with other attack methods are not fair if you do not guarantee the same level of image quality. For example, in case of BIM [22] the visual quality is much higher (see Fig. 6). Overall when comparing with prior art it should be indicated the image quality of the attacked images so that all methods are compared in a fair way.
> >
> > Finally, in the ablation study I would have expected a comparison with a simple baseline where the attack is performed in step 6. As you said this is a ''trivial solution'', however it is important to show that this trivial solution is not working better than the proposed one.
> >
> > I would be grateful if you could clarify these points.

---

> > > ### Author Response · Authors · 2021-08-27
> > > **Further response to Reviewer LYp3**
> > >
> > > Thanks for your meaningful comments. We try our best to address your further concerns as follows.
> > > ***
> > > Q10: The visible distortions caused by Moiré Attack (MA).
> > >
> > > A10: First, the moiré pattern generated by shooting on LCD is affected by several factors such as background light intensity and shooting angles. Due to the internal process of camera (step 2 in Section 3.3), the image perceived by the sensor is darker. We use *iPhone 12 rear camera* to take pictures on the *AOCU27P1U 4K 3840\*2160dpi* monitor and *MacBook Air 2020 2560\*1600dpi* monitor. The comparisons are shown in Figure 10 (c) (d) (https://www.dropbox.com/s/vbinjg89ldefh9p/Figure10.png?dl=0). We can see that the images shot on LCD are darker in general, especially when the brightness of the monitor is higher.
> > >
> > > In addition, *Challenge on Image Demoiréing* [1][2] uses an official dataset called **LCDMoiré** [2] for benchmarking example-based image demoiréing. We strictly follow the same procedure in our simulation of moiré pattern. Our generated images are also consistent with the ones in the literature [2][9], as shown in Figure 10 (a) (b) (https://www.dropbox.com/s/vbinjg89ldefh9p/Figure10.png?dl=0), we display some images in **LCDMoiré** and their clean counterpart. The synthesis images are darker and distorted to some degree compared with the original ones.
> > > ***
> > > Q11: The fairness of the comparison between MA and other attack methods.
> > >
> > > A11: As we mentioned in the above Q4\&A4, our proposed MA is like a combination of physical attack and digital attack. The common digital attack aims to mislead CNNs with imperceptible noise while physical attack utilizes natural phenomena to deliver the attack (so its distortion is inevitably distinct). The proposed MA takes advantage of the two: it has the comparable success rate with digital attack and more reasonable distortion in human perception compared with traditional physical attack (see the generated adversarial examples by AdvPatch [3] and AdvCam [4] in Figure 8 (https://www.dropbox.com/s/xr7fgbvv1ntumyn/Figure.png?dl=0). In other words, it does have a distortion, but the distortion is reasonable in the specific condition (shooting on the screen) and seldom raises human attention. Furthermore, as for the visible quality of MA and BIM in Figure 6, we do conduct the experiment to exclude the factors of the background light intensity and the rotation angle that may affect the attack results. Table 1 shows that the victim CNN is very robust against transformations including rotating and dimming + rotating, so the darkness of the adversarial example is just an unavoidable by-product of practical shooting process but seldom contributes to its adversarial property.
> > > ***
> > > Q12: An ablation study to compare MA to a baseline with noise added in step 6.
> > >
> > > A12: In the perspective of the practical rationality of the proposed MA, we have explained in the above Q7&A7 that we optimize the noise to simulate the sensor noise carried by the camera in the specific step 5 which is meaningful because of its unavoidability in the real process.
> > >
> > > Following your suggestion, we conduct additional experiments to compare untargeted MA to a simple baseline with noise added in step 6 (Moiré + BIM). The success rate before and after JPEG compression and resampling are shown in Table 7 and Table 8. We can see that in Moiré + BIM, the noise is easier to be removed by compressing or resampling the adversarial example. In this aspect, the proposed MA is more robust under image transformations.
> > > ***
> > > **Table 7: The success rate of untargeted MA and Moiré + BIM under JPEG compression.** The noise budget $\epsilon$ is 4.
> > >
> > > |  &emsp; Attack   | Before JPEG compression | qf=20 | qf=40 |
> > > |:-------------:|:-----------------------:|:-----:|:-----:|
> > > |      MA     |          **1.000**          | **0.938** | **0.949** |
> > > | Moiré + BIM |          0.998          | 0.857 | 0.902 |
> > > ***
> > > **Table 8: The success rate of untargeted MA and Moiré + BIM under resampling.** The noise budget $\epsilon$ is 4. The original size of the adversarial example is 299 * 299 * 3, the third and fourth columns signify the success rate when the adversarial example is resampling to the specific size.
> > >
> > > | &emsp;Attack   | Before resampling | [224, 224, 3] | [184, 184, 3] |
> > > |:-----------:|:-----------------:|:----------:|:----------:|
> > > |      MA     |       **1.000**       |    **0.962**   |    **0.946**   |
> > > | Moiré + BIM |       0.998       |    0.917   |    0.898   |
> > > ***
> > > [1] Yuan, Shanxin, et al. "Ntire 2020 challenge on image demoiréing: Methods and results." Proceedings of the IEEE/CVF Conference on Computer Vision and Pattern Recognition Workshops. 2020.
> > >
> > > [2] Yuan, Shanxin, et al. "Aim 2019 challenge on image demoiréing: Dataset and study." 2019 IEEE/CVF International Conference on Computer Vision Workshop (ICCVW). IEEE, 2019.
> > >
> > > [3] Brown, Tom B., et al. "Adversarial patch." arXiv preprint arXiv:1712.09665 (2017).
> > >
> > > [4] Duan, Ranjie, et al. "Adversarial camouflage: Hiding physical-world attacks with natural styles." Proceedings of the IEEE/CVF conference on computer vision and pattern recognition. 2020.
> > >
> > > [9] Liu, Bolin, Xiao Shu, and Xiaolin Wu. "Demoir\'eing of Camera-Captured Screen Images Using Deep Convolutional Neural Network." arXiv preprint arXiv:1804.03809 (2018).

---

### Official Review · Reviewer_zVAp · 2021-07-16

**Rating:** 7
**Confidence:** 4

**Summary:**

This paper presents an interesting new deep attack method, which utilizes the morie effect caused by digital image processing. It mimics the shooting process of digital devices and generates the physical-world morie pattern, which is then added to the targeted images to attack a given deep model. Extensive experiments are conducted to support the proposed method.

**Limitations And Societal Impact:**

No concern was found. This paper should have positive social impact.

**Main Review:**

Pros:
-	Strong and clear motivation. The authors first analyze the special phenomenon in digital image processing, the morie effect, which could cause unnotic3ed security threats to DNNs. And then based on this phenomenon, it develops the morie attack methods. The method is well motivated.
-	Solid technical approach. The proposed morie attack can mimic the shooting process of digital devices and generate the physical-world morie pattern by the proposed adversarial loss. The obtained morie pattern can then be added to images to attack a given deep model.
-	State of the art experimental results. Their simple approach yields significant attack performance on ImageNet and various DNN models.

Cons:
-	The generated adversarial examples are not so realistic compared with images with normal morie pattern.
-	There should be more explanation in the title of Figure 4 to make the figure self-explainable.
-	(minor) There are several typos, that should be revised. For example: in line 135, "a physical threat to DNNa that ...". The presentation can be improved.


**Time Spent Reviewing:**

2

---

> ### Author Response · Authors · 2021-08-10
> **Response to Reviewer zVAp**
>
> Thanks for your thoughtful comments. We provide the following responses to your concerns.
> ***
> Q1: The generated adversarial examples with moiré pattern are not so realistic as the normal one.
>
> A1: We agree with you that it may be a little bit weird when first observing the generated examples with moiré patterns, but the phenomenon does exist in the real process of shooting images on the LCD monitor in practice. In fact, the feeling of abnormality is caused by the comparison with the clean original image, while we are taking images on LCD monitors. Maybe you can try now and use your mobile phone to take a photo of the screen in front of you. You will find that the obtained image will be distorted and darker compared with the original one. It is also consistent with the analysis in lines 185 - 188 that the unavoidable real process of resampling the pixels into subpixels indirectly causes that distortion.
>
> Furthermore, *Challenge on Image Demoiréing* [1][2] uses an official dataset called **LCDMoiré** [2] for benchmarking example-based image demoiréing. We strictly follow the same way to generate **LCDMoiré** in our generation of moiré pattern. Our generated images are also consistent with the ones in their paper, that is, the synthesis images are darker and distorted in some degree compared with the original ones.
>
> ***
> Q2: Figure 4 needs more explanation to make it self-explainable.
>
> A2: Thanks for your suggestion. We rewrite the title of Figure 4 as follows: The pipeline of Moiré attack (MA). To generate the moiré pattern, the input image is first fed into the yellow module which mimics the shooting process of taking photos on LCD monitors in which an iterative sensor noise $\delta$ is added in the step of Bayer CFA. The output image with moiré pattern is then fed into a CNN to make a prediction where backpropagation is conducted through the CNN to iterate the sensor noise $\delta$ to mislead the CNN to make wrong predictions.
>
> ***
> Q3: Minor typos and language errors.
>
> A3: Thanks for pointing them out. We will fix them in the revision.
>
> ***
> [1] Yuan, Shanxin, et al. "Ntire 2020 challenge on image demoiréing: Methods and results." IEEE/CVF Conference on Computer Vision and Pattern Recognition Workshops. 2020.
>
> [2] Yuan, Shanxin, et al. "Aim 2019 challenge on image demoiréing: Dataset and study." IEEE/CVF International Conference on Computer Vision Workshop. 2019.

---

### Decision · Program_Chairs · 2021-09-27

**Decision:**

Accept (Poster)

**Comment:**

The paper presents a novel kind of attack: Moire attack. It is inspired by that there will be Moire effect when shooting images on the LCD monitors. Although the Moire effect is perceptible by human eyes, it is very hard to distinguish between images with different Moire effects. Moire attack therefore is hard to recognize and will be harmful. Current algorithms are not robust to Moire attack. The authors are calling for our attention to this attack. We suggest the author carefully merge the rebuttals in the final version.